# Suppression of hTAS2R16 Signaling by Umami Substances

**DOI:** 10.3390/ijms21197045

**Published:** 2020-09-24

**Authors:** Mee-Ra Rhyu, Yiseul Kim, Takumi Misaka

**Affiliations:** 1Division of Functional Food Research, Korea Food Research Institute, Jeollabuk-do 55365, Korea; kimys@kfri.re.kr; 2Department of Applied Biological Chemistry, Graduate School of Agricultural and Life Sciences, The University of Tokyo, Tokyo 113-8657, Japan; amisaka@g.ecc.u-tokyo.ac.jp

**Keywords:** umami, bitter suppression, calcium signaling, hTAS2R16 N96T, hTAS2R16 P44T, GPCR, *TAS2R16*

## Abstract

Interaction between umami and bitter taste has long been observed in human sensory studies and in neural responses in animal models, however, the molecular mechanism for their action has not been delineated. Humans detect diverse bitter compounds using 25-30 members of the type 2 taste receptor (TAS2R) family of G protein-coupled receptor. In this study, we investigated the putative mechanism of antagonism by umami substances using HEK293T cells expressing hTAS2R16 and two known probenecid-insensitive mutant receptors, hTAS2R16 N96T and P44T. In wild type receptor, Glu-Glu, inosine monophosphate (IMP), and l-theanine behave as partial insurmountable antagonists, and monosodium glutamate (MSG) acts as a surmountable antagonist in comparison with probenecid as a full insurmountable antagonist. The synergism with IMP of umami substances still stands in the suppression of hTAS2R16 signaling. In mutagenesis analysis, we found that Glu-Glu, MSG, and l-theanine share at least one critical binding site on N96 and P44 with probenecid. These results provide the first evidence for a direct binding of umami substances to the hTAS2R16 through the probenecid binding pocket on the receptor, resulting in the suppression of bitterness.

## 1. Introduction

The human gustatory system recognizes five primary taste qualities: sweet, bitter, umami, sour, and salty. While ion channels serve as sensors for detecting sour and salty taste, G-protein-coupled receptors (GPCRs) serve as receptors for detecting sweet, umami, and bitter taste stimuli [1,2,3,4,5]. Recent progress in the identification and characterization of specific taste receptors and their downstream signaling cascades involved in the taste transduction has made it easier to make an objective quantification or qualification of each taste quality [6]. However, an influence of a unique taste quality to enhance or suppress taste intensity of one or more taste qualities has been observed in psychophysical [5,7,8] and neural studies [9,10,11]. For instance, oligopeptides are generally produced by enzymatic hydrolysis of diverse food proteins during aging or fermentation, playing an important role in imparting a more complete flavor of final food products [12,13,14,15,16]. Some peptides themselves elicit bitter, umami, and salty taste [17], while other small, di-, or tri-, peptides can modify the taste of the food as zwitterions in aqueous phase [18]. Another important consideration of peptides is their taste-modulating ability. In particular, umami peptides suppress the strong bitterness of the peptic hydrolysate of soybean protein that are formed during the course of the plastein synthesis reaction [19,20,21,22,23,24]. It is still an open question as to how this interaction between two different taste qualities, umami and bitter, are recognized at the taste receptor level.

The receptors for bitter compounds are defined as the type 2 taste receptor (TAS2R) family of GPCR. Genetic studies suggest that humans use at least 25 members of the TAS2R family to detect thousands of structurally diverse bitter compounds [25,26,27,28]. Because some commercially available high-potency non caloric sweeteners including saccharine or acesulfame K leave bitter a aftertaste in humans, much effort has been invested in delineating the underlying molecular mechanism for this interaction [29]. The identification of hTAS2R31 as a selective receptor to saccharine or acesulfame K enabled a deliberately targeted screening to find novel antagonists of bitter taste [30]. In spite of the high demand of food industry for such novel antagonists of bitter taste, only two selective antagonists for the TAS2R family have been identified at present [31,32]. The first small molecule antagonist, 4-(2,2,3-trimethylcyclopentyl) butanoic acid (GIV3727), was identified from around 18,000 compounds screened. It most likely acts as an orthostaric, insurmountable antagonist and can also inhibit hTAS2R43, another receptor selective for saccharine or acesulfame K [32]. The other selective antagonist, *ρ*-(dipropylsulfamoyl)benzoic acid (probenecid; commonly used to prevent the efflux of calcium-sensitive fluorescent dyes [33]), acts as a full insurmountable antagonist on hTAS2R16. It selectively inhibits the binding of bitter ligand β-glucopyranosides including that of salicin, arbutin, and phenyl-β-d-glucopyranoside [26,31]. Analysis of a point mutant receptor disclosed two binding sites for probenecid at amino acid positions N96 in transmembrane 3 and P44 in transmembrane 2 [31]. Our previous investigations conducted with umami peptides using cells expressing hTAS2R16 indicate that umami peptide selectively suppresses human bitter taste receptor signaling [34]. The suppression of receptor signaling of caffeine has also been observed in cells expressing hTAS2R43 and hTAS2R46. Umami-active peptide fractions naturally generated during maturation of Korean soy sauce inhibit the receptor specific for caffeine [35]. These studies support the notions that umami peptides can produce receptor-mediated inhibition of bitter ligand binding on human bitter taste receptors. Similar suppression of bitter taste was observed not only by peptides but also by other umami substances such as l-glutamate and 5′-ribonucleotides in both human sensory evaluation studies [7,36,37] and in neural responses in an animal model [10]. At present, it is not clear if l-amino acids and 5′-ribonucleotides also suppress bitter taste by a similar mechanism similar to that of umami peptides and if this mechanism is different from described for probenecid on hTAS2R16. A strong synergism between umami substances is a unique feature that distinguishes it from other primary taste qualities [38,39,40]. However, in addition to inosine monophosphate (IMP) and monosodium glutamate (MSG), this synergism is also possessed by Glu-Glu and l-theanine (γ-glutamylethylamide), a principal umami compound found in green tea [15,41]. Therefore, it is imperative to investigate the synergism with IMP for the Glu-Glu and l-theanine-induced inhibition of bitter taste using the cells expressing hTAS2R16. Recently, it has been elucidated that bitter taste receptors are also actually expressed in the extra oral tissues, such as intestine, brain, bladder, and lower and upper respiratory tract [42]. Furthermore, bitter taste receptors have been identified as potentially important therapeutic targets for diseases associated with the above extra oral tissues. The detailed molecular mechanisms involved in regulating the activity of bitter taste receptor, therefore, can provide useful information for the future design and development of therapeutic intervention.

In this paper, we investigated the effects of MSG, IMP, and l-theanine on the binding of salicin to the bitter taste receptor in cells expressing hTAS2R16. The inhibitory potency of MSG, IMP, and l-theanine was compared with that of Glu-Glu, an umami peptide. We investigated the putative mechanism of antagonism of the hTAS2R16 by umami substances and if the synergism in the suppression of receptor signaling is observed both with di-peptide and non-peptide umami substances. Using two known probenecid-insensitive hTAS2R16 mutant receptors, we determined the potential binding sites of umami substances. Our results demonstrated that Glu-Glu, IMP, and l-theanine behaved as partial insurmountable antagonists and that MSG behaved as a surmountable antagonist. These results demonstrated that the synergism between umami substances in the suppression of hTAS2R16 signaling is a common feature of umami substances. Umami substances directly bind to N96 and/or P44, critical binding sites for probenecid in hTAS2R16. We conclude that interactions between umami and bitter taste occur at the receptor level.

## 2. Results and Discussion

### 2.1. Effect of MSG, IMP, and l-Theanine on Ligand Binding to the hTAS2R16

We previously showed that umami peptide and peptide mixture suppress human bitter taste receptor signaling in cells expressing hTAS2R16, hTAS2R43, or hTAS2R46 [34,35]. We first verified if non-peptide umami substances affect the downstream signaling mechanism, such as changes in intracellular Ca^2+^ of the human bitter taste receptor. To test this, we monitored temporal changes in intracellular Ca^2+^ in HEK293T cells expressing hTAS2R16 in response to salicin in the absence and presence of umami substances MSG, IMP, and l-theanine. An umami peptide Glu-Glu and a tasteless peptide Gly-Gly were used as the basis for comparison (Figure 1A). The increase in fluorescence intensity ratio (F_340_/F_380_) with 10 mM salicin was expressed as 100. As shown in Figure 1B,C, all umami substances tested—MSG (10 and 80 mM), IMP (10 and 20 mM), l-theanine (10 and 80 mM), and Glu-Glu (6 mM)—elicited a concentration-dependent reduction of intracellular Ca^2+^ influx evoked by salicin. A similar concentration-dependent reduction of intracellular Ca^2+^ was induced by probenecid (0.1 and 0.3 mM), a TAS2R16-specific antagonist [31], but not by Gly-Gly (6 mM). MSG, IMP, l-theanine, Glu-Glu, and Gly-Gly by themselves did not alter salicin response at the concentrations used in these experiments (Figure 1B). These data suggest that the suppression of bitter receptor signaling is not limited to umami peptides but may be a common property of umami tasting substances. In addition to salicin, hTAS2R16 also recognizes bitter sugars such as phenyl β-d-glucopyranoside and arbutin, a glycosylated hydroquinone extracted from the bearberry plant in the genus *Arctostaphylos* [30]. Cells expressing TAS2R16 respond to an increase in Ca^2+^ when stimulated with arbutin (Figure 1D) and phenyl β-d-glucopyranoside (Figure 1E) that was also inhibited by probenecid. Relative to the Ca^2+^ response of arbutin or phenyl β-D-glucopyranoside alone, in mixtures containing MSG (80 mM), IMP (20 mM), l-theanine (80 mM), and Glu-Glu (6 mM), the Ca^2+^ response was significantly inhibited, but not by Gly-Gly (6 mM). These results indicate that umami substances, irrespective of their structure, interact with hTAS2R16 and inhibit increase in cell Ca^2+^ induced by its respective agonists. 

### 2.2. The Inhibitory Potency of Umami Substances on Salicin Binding to the hTAS2R16

To assess the inhibitory potency of the umami compounds on salicin binding to hTAS2R16, we determined the half maximal inhibitory concentrations (IC_50_) by adding increasing concentration of MSG, IMP, l-theanine, and Glu-Glu using a fixed salicin concentration (10 mM) in HEK293T cells expressing hTAS2R16, and compared them with that of probenecid (Figure 2A). Increasing concentration of Glu-Glu, MSG, IMP, and l-theanine inhibited the increase in intracellular Ca^2+^ in a dose-dependent manner. MSG, IMP, and l-theanine inhibition was observed between 20 and 100 mM, a concentration ranges significantly higher than that of Glu-Glu (6–20 mM) or probenecid (0.03–1.0 mM) (Figure 2A). The calculated IC_50_ of probenecid was similar to the reported IC_50_ of 0.29 mM in the presence of 3 mM salicin [31], and Glu-Glu showed the lowest IC_50_ value, followed by IMP of the umami substances tested. MSG and l-theanine inhibited the Ca^2+^ response at the highest concentrations and were equally effective with the mean IC_50_ values of 51 and 62 mM, respectively (Figure 2B). In contrast to the concentrations that suppress salicin response, in human sensory evaluation studies, MSG (30 mg%) showed around 10 times lower umami taste threshold than Glu-Glu (150–300 mg%) [21,40,43]. Similarly, IMP (0.35) was found to be far weaker than that of MSG (1.0) in enhancing umami taste intensity [40]. In terms of detection threshold, l-theanine (25 mM) [41] was found to be roughly 36 times lower than that of MSG (0.7 mM) [5]. These results indicate that the potency of umami substances to suppress hTAS2R activation by salicin cannot be directly correlated with their umami taste intensity.

### 2.3. Mechanism of Antagonism of the hTAS2R16 by Umami Substances

Receptor antagonists can be classified as surmountable and insurmountable antagonists [44]: the former produce parallel rightward shifts of agonist concentration–response curve with no changes in the maximal agonist response, whereas the latter also decrease the maximal agonist response. We investigated the putative pattern of antagonism of salicin concentration response using a fixed concentration of MSG (10, 30, 80, and 100 mM), IMP (1, 10, 20, and 30 mM), l-theanine (10, 30, 80, and 100 mM), and Glu-Glu (1, 3, 6, and 10 mM) to test if they act as surmountable or insurmountable antagonists on salicin binding to hTAS2R16 (Figure 3). To assess the potency of the antagonists, we obtained the half maximal effective concentration (EC_50_) of each graded concentration of the substances (Table 1). The estimated EC_50_ of salicin was 0.89-1.06, which was close to the reported value of 1.2–1.4 mM [26,31]. Consistent with previous studies, increasing the concentration of probenecid (0.1, 0.3, and 1 mM) did not change the EC_50_ of salicin (Table 1), in which the maximal response of salicin alone was suppressed by 43% at 0.3 mM before the rightward concentration-dependent shift of the response, which is typically indicative of a full insurmountable antagonism (Figure 3A) [45]. Increasing concentration of Glu-Glu, IMP, and l-theanine increased EC_50_ by 2–3-fold, and the maximal response of salicin alone was gradually decreased by 61%, 50%, and 55% at the highest concentration of Glu-Glu (10 mM), IMP (30 mM), and l-theanine (100 mM), respectively, with a gradual rightward shift in the salicin concentration–response curve (Figure 3B,D,E and Table 1). These results indicate that Glu-Glu, IMP, and l-theanine behave as partial insurmountable antagonists at the hTAS2R16 [45]. In contrast with Glu-Glu, IMP, and l-theanine, increasing concentration of MSG produced parallel rightward shifts of the salicin dose–response curve with no change in the maximal response (Figure 3C), and 8.5-fold increases of the EC_50_ of salicin at the highest concentration of MSG (100 mM) (Table 1). These results indicate that MSG behaves as a surmountable antagonist of hTAS2R16 [45]. These results further support that peptide and non-peptide umami substances including Glu-Glu, MSG, IMP, and l-theanine, irrespective of their structure, likely behave as antagonists at the hTAS2R16 with their own mechanisms.

### 2.4. Synergism between Umami Substances on the Suppression of Bitterness

The strong synergistic interaction between two classes of umami substances, l-glutamate and 5′ ribonucleotides, is a distinct characteristic that distinguishes umami taste quality from other primary taste qualities [38]. Synergistic interaction with IMP has also been demonstrated for Glu-Glu in an earlier human study [15] and for l-theanine in both human sensory evaluation and mouse gustatory nerve recordings [41]. The molecular mechanism for the taste-enhancing synergism between two umami substances, glutamate, and IMP has been explained as a cooperative binding of IMP adjacent to glutamate binding site, stabilizing the closed form of the Venus flytrap domain of T1R1 of T1R1/T1R3, an umami taste receptor, through electrostatic interactions [46]. Since umami substances such as Glu-Glu, IMP, MSG, and l-theanine may bind to potential sites on hTAS2R16 (Figure 3) that are different from the orthostaric agonist binding site, we further determined whether the synergism was still observed in the suppression of receptor signaling in cells expressing hTAS2R16. Because IMP itself stimulates the hTAS2R16 at higher concentrations than 10 mM (Figure 3), we calculated changes by IMP of the IC_50_ values of Glu-Glu, MSG, and l-theanine in the presence of 10 mM salicin at lower concentrations than 10 mM IMP. Increasing concentration of IMP produced little changes in the IC_50_ of Glu-Glu in the presence of salicin (Figure 4A and Table 2), suggesting that the synergistic interaction between IMP and Glu-Glu does not lead to the suppression of salicin binding. However, the MSG (1–300 mM) concentration curve in the presence of 10 mM salicin was shifted leftward upon the addition of graded concentration of IMP (1, 5, and 10 mM) (Figure 4B). The estimated IC_50_ decreased gradually from the normalized value of 1.0 in the absence of IMP to around 0.93, 0.77, and 0.3 in its presence (Table 2). By contrast, the l-theanine concentration curve in the presence of 10 mM salicin produced a downward shift of the maximal response upon adding graded concentration of IMP (1, 3, and 10 mM) with a small leftward shift (Figure 4C). The IC_50_ decreased from 1.0 (in the absence of IMP) to 0.9 upon adding 10 mM IMP (Table 2). Although at present the molecular mechanism of synergism between umami substances is not clear, our results provide evidence that some umami substances, such as MSG and l-theanine, produce similar synergistic effects with IMP by altering salicin binding to hTAS2R16 by a separate mechanism, whereas Glu-Glu does not.

### 2.5. Point Mutant Analysis Using Two Known Probenecid-Insensitive Mutants, hTAS2R16 N96T and P44T, for the Potential Binding Sites of Umami Substances

Probenecid, a selective antagonist on hTAS2R16, binds to the amino acid positions N96 in transmembrane 3 and P44 in transmembrane 2 [31]. It is interesting to note that Glu-Glu, MSG, IMP, and l-theanine share these binding sites with probenecid to inhibit ligand binding to the receptor. We thus performed point mutant analysis using two known probenecid-insensitive mutant receptors, hTAS2R16 N96T and P44T, to define the potential binding positions of four umami substances. Salicin (0.003–30 mM) induced intracellular Ca^2+^ response in all cells expressing wild-type (WT) receptor and the two mutants N96T and P44T. As expected, a significant attenuation of salicin response by pretreatment with probenecid was observed only in cells expressing the WT receptor but not in cells expressing the two mutants (Figure 5). As with probenecid, pretreatment with l-theanine (80 mM) significantly attenuated salicin response only in WT receptor and not in cells expressing the two N96T and P44T mutant receptors. This indicates that l-theanine behaves like probenecid through binding to the amino acid residues N96 and P44, which are critical binding sites for probenecid (Figure 5). In contrast, only the N96T mutant receptor lost its inhibitory action for salicin response (0.003–30 mM) by Glu-Glu. This suggests that Glu-Glu shares a binding site at N96 with probenecid. In contrast, MSG altered salicin activity in P44T mutant, suggesting a separate binding position for Glu-Glu (Figure 5). Our analysis using N96 and P44 mutant suggests that IMP binds to different amino acid residues than probenecid to antagonize the hTAS2R16 (Figure 5 and Table 3). These data provide strong evidence that umami substances directly bind to the bitter taste receptor hTAS2R16 and that Glu-Glu, MSG, and l-theanine bind to the probenecid binding pocket on the receptor.

## 3. Materials and Methods

### 3.1. Materials

Salicin, probenecid, MSG (l-glutamic acid monosodium salt hydrate), IMP (inosine 5′-monophosphate disodium salt hydrate), and l-theanine were purchased from Sigma-Aldrich (St. Louis, MO, USA). The dipeptide Glu-Glu was synthesized from Lugen Sci (Seoul, Korea).

### 3.2. Cell Culture and Transfection

HEK293T cells were purchased from American Type Culture Collection (ATCC; Manassas, VA, USA) and cultured as described previously [47]. The plasmids hTAS2R16 and G16αgust44 were constructed as described previously [48,49]. G16αgust44 and hTAS2R16 were cloned into a cytomegalovirus promoter-based vector and expressed constitutively. Point mutants of N96T and P44T on hTAS2R16 clone and DNA sequencings were performed by Macrogen (Geumcheon-gu, Seoul, Korea). The hTAS2R16 or point mutant expression plasmid was co-transfected with G16αgust44 expression plasmid (4:1) into HEK293T cells using Lipofectamine 2000 (Invitrogen, Carlsbad, CA, USA) [50].

### 3.3. Measurement of Cellular Ca^2+^ Responses in HEK293 Cells Expressing Wild-Type and Mutant hTAS2R16 by Cell-Based Assay

The umami substances-induced changes in the cytosolic Ca^2+^ concentration were measured using a FlexStation III microplate reader (Molecular Devices, Sunnyvale, CA, USA). HEK293T cells transfected with hTAS2R16 were seeded onto 96-well black-wall CellBind surface plates (Corning, NY, USA) for 18–24 h before use. Following this, cells were washed with buffer (130 mM NaCl, 10 mM glucose, 5 mM KCl, 2 mM CaCl_2_, 1.2 mM MgCl_2_, and 10 mM 4-(2-hydroxyethyl)-1-piperazine ethanesulfonic acid (HEPES) (pH 7.4)). Because acidic pH was demonstrated as one of the critical factors responsible for the bitterness-masking effect in cells expressing bitter taste receptor, we maintained a pH of 7.4–7.25 in all assay systems with 100 mM HEPES in the buffer to exclude acidic pH effect, as was performed in a previous study [51]. The cells were incubated in the dark for 30 min at 37 °C, and then for 15 min at 27 °C in assay buffer consisting of Calcium-4 (FLIPR Calcium 4 Assay Kit, Molecular Devices) and probenecid or the umami compounds. Following treatment, the fluorescence intensity (excitation, 486 nm; emission, 525 nm) of the cell was measured. The relative changes in cell Ca^2+^ were plotted as ΔF/F_0_ on the y-axis, where ΔF is the change in Calcium-4 fluorescence intensity at each time point and F_0_ is the fluorescence intensity before treatment. The relative response induced by 10 mM salicin was assigned a value of 100.

### 3.4. Measurement of Cellular Ca^2+^ Responses in HEK293T Cells Expressing hTAS2R by Calcium Imaging Analysis

Cells were seeded onto 96-well black-wall imaging plates (BD Falcon Labware) for 18–26 h prior to their use in an experiment. After this time, the plates were changed with 100 mM HEPES (pH 7.4) including 130 mM NaCl, 10 mM glucose, 5 mM KCl, 2 mM CaCl_2_, and 1.2 mM MgCl_2_, and then loaded with 5 μM Fura-2AM (Invitrogen), a Ca^2+^ indicator, in assay buffer for 30 min at 37 °C. The cells were rinsed with the buffer, incubated in probenecid or sample for 30 min, and then treated with solution containing the ligand. The Fura-2AM-induced fluorescence intensity (excitation at 340 and 380 nm) was measured at emission wavelength of 510 nm using a computer-controlled filter changer (Lambda DG4; Sutter, San Rafael, CA, USA), an Andor Luca CCD camera (Andor Technology, Belfast, UK), and an inverted fluorescence microscope (IX-71; Olympus, Tokyo, Japan). Images were obtained at 3-s intervals and analyzed using MetaFluor software (Molecular Devices, Sunnyvale, CA, USA).

### 3.5. Statistical Analysis

All statistical analysis and dose–effect curves were analyzed by GraphPad Prism 5.0 (GraphPad Software Inc., San Diego, CA, USA). Data are presented as means ± standard error of the mean (SEM).

## 4. Conclusions

In this paper, we determined the molecular mechanism of antagonism by umami substances on hTAS2R16 human bitter taste receptor. Our data suggest that four tested umami compounds Glu-Glu, IMP, l-theanine, and MSG behave as partial insurmountable or surmountable antagonists. This is in contrast to probenecid that behaves as a full insurmountable antagonist. Although the synergism between umami substances has been explained as a cooperative ligand-binding to stabilize umami receptor T1R1/T1R3, it still stands in the suppression of hTAS2R16 signaling. Point mutant analysis using two known probenecid-insensitive mutant receptors provides new evidence that umami substances directly bind to N96 and/or P44, critical binding sites for probenecid in hTAS2R16. Our results further indicate that interaction between umami and bitter taste occur at the receptor level.

## Figures and Tables

**Figure 1 ijms-21-07045-f001:**
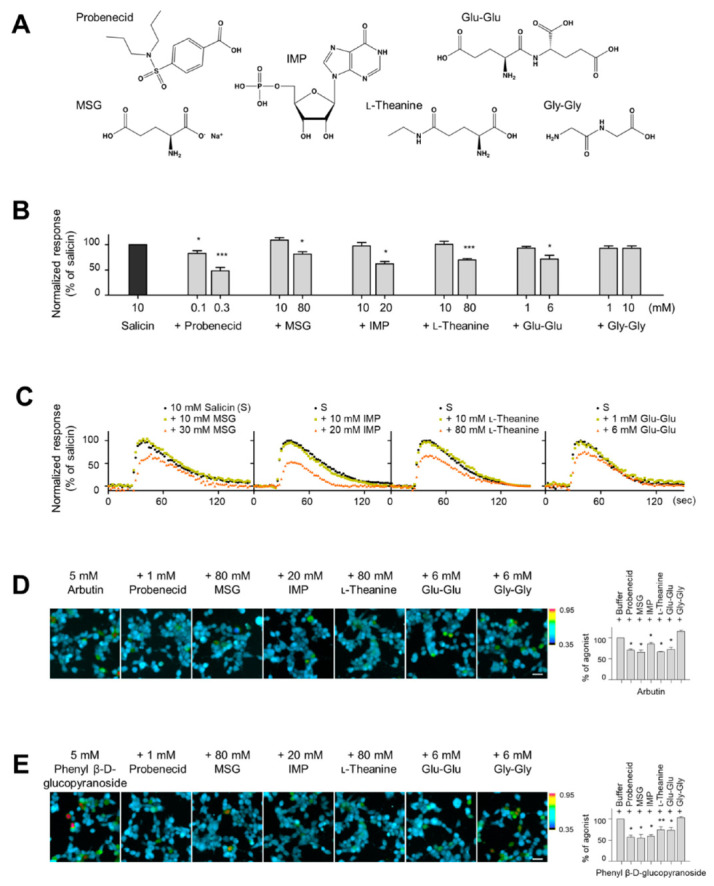
Effect of monosodium glutamate (MSG), inosine monophosphate (IMP), and l-theanine on bitter ligand salicin binding in cells expressing hTAS2R16. (**A**) Chemical structures of tested compounds. Summarized data (**B**) and representative traces (**C**) showing responses of HEK293T cells co-expressing hTAS2R16 and Gα16gust44 to 10 mM salicin in the absence and presence of umami substances in a cell-based assay. (**D****,E**) Representative ratiometric images of fura-2-loaded hTAS2R16-expressing cells in response to the other ligands, 5 mM arbutin (**D**) or phenyl β-D-glucopyranoside (**E**), in the presence of umami substances. The data were normalized as percentages to the maximal signal obtained with the ligands. The color scale indicates the F_340_/F_380_ ratio. Scale bar: 11.04 μm. Data are presented as means ± standard error of the mean (SEM) of at least three separate experiments. * *p* < 0.05, ** *p* < 0.01, and *** *p* < 0.001 vs. the 0 mM sample group by Student’s *t*-test.

**Figure 2 ijms-21-07045-f002:**
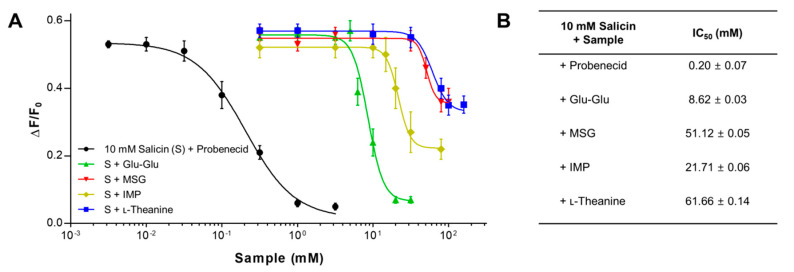
The inhibitory potency of umami substances on bitter ligand salicin binding to the hTAS2R16. (**A**) Concentration–response profiles obtained from pretreatment of umami substances in the presence of 10 mM salicin in HEK293T cells expressing hTAS2R16. The data were normalized as percentages to the maximal signal obtained with the ligand. Data were presented as means ± SEM of at least three separate experiments performed and fitted in GraphPad Prism. (**B**) IC_50_ values of umami substances in the presence of 10 mM salicin.

**Figure 3 ijms-21-07045-f003:**
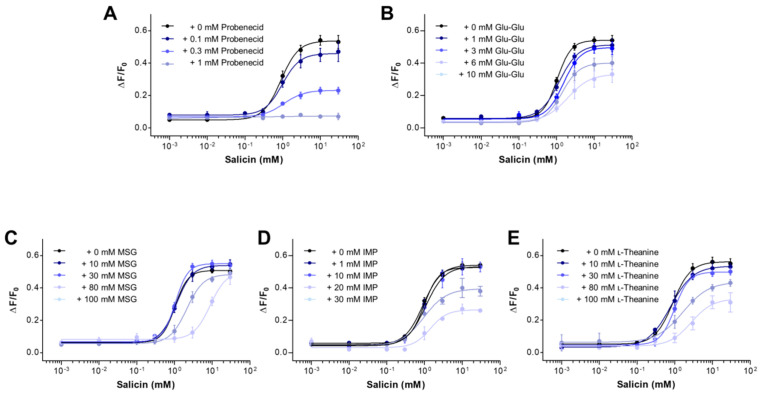
Mechanism of antagonism of the umami substances in HEK293T cells expressing hTAS2R16. Concentration–response profiles of the intracellular Ca^2+^ level induced by salicin (0.001–30 mM) by adding increasing concentration of probenecid (**A**), Glu-Glu (**B**), MSG (**C**), IMP (**D**), and l-theanine (**E**) in HEK293T cells expressing hTAS2R16. Data are presented as means ± SEM of at least three separate experiments performed and fitted in GraphPad Prism.

**Figure 4 ijms-21-07045-f004:**
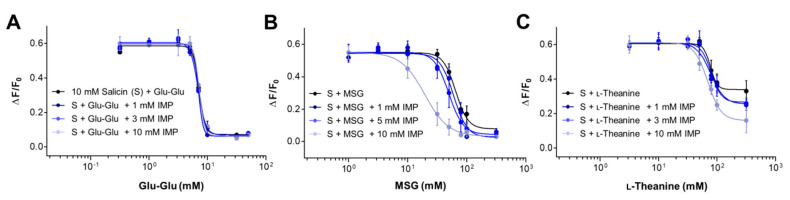
The synergistic effect with IMP of umami substances in the suppression of bitterness. Concentration–response profiles obtained from pretreatment of Glu-Glu (**A**), MSG (**B**), and l-theanine (**C**) in the presence of 10 mM salicin by adding graded concentration of IMP in HEK293T cells expressing hTAS2R16. Data are presented as means ± SEM of at least three separate experiments performed and fitted in GraphPad Prism.

**Figure 5 ijms-21-07045-f005:**
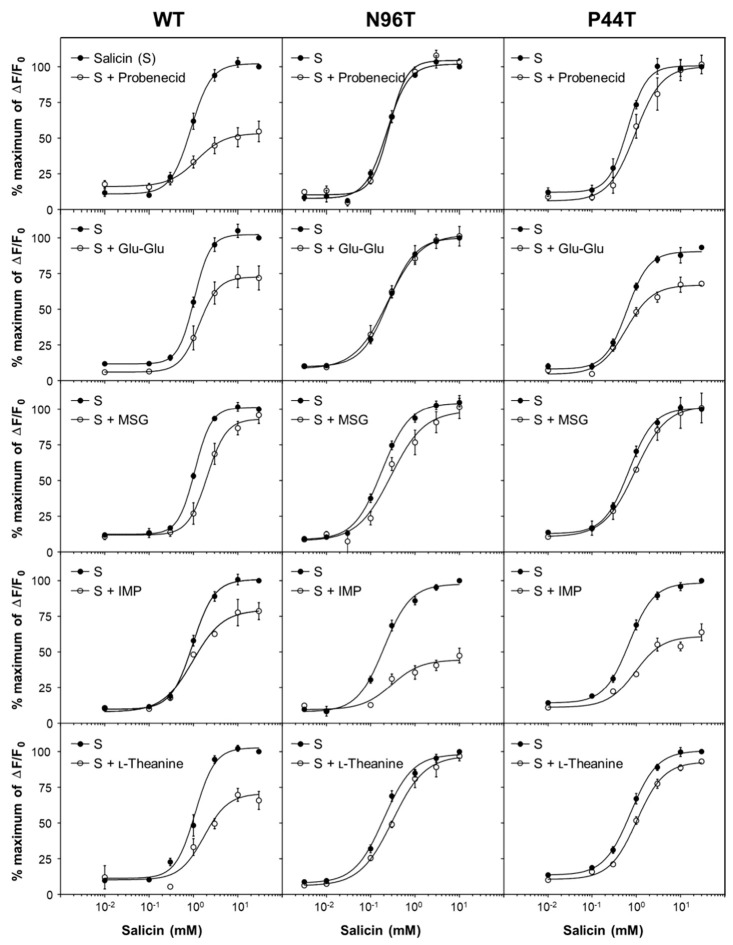
Concentration–response profiles obtained from pretreatment of probenecid (0.3 mM), Glu-Glu (6 mM), MSG (80 mM), IMP (20 mM), and l-theanine (80 mM) in the presence of salicin (0.001–30 mM) in wild-type hTAS2R16 and two known probenecid-insensitive mutants, N96T and P44T. The data were normalized as percentages to the maximal signal obtained with the ligand. Data are presented as means ± SEM of at least three separate experiments performed and fitted in GraphPad Prism.

**Table 1 ijms-21-07045-t001:** Changes of EC_50_ values and maximal responses of salicin by preatreatment of increasing concentrations of umami substances in cells expressing hTAS2R16.

Salicin (0–30 mM) + Sample (mM)		EC_50_ (mM)	Maximal Response (ΔF/F_0_)
+ Probenecid	0	0.90 ± 0.06 ^†^	0.54
	0.1	0.98 ± 0.10	0.47
	0.3	1.07 ± 0.11	0.23
	1	NE ^‡^	0.07
+ Glu-Glu	0	1.04 ± 0.03	0.54
	1	1.18 ± 0.05	0.51
	3	1.53 ± 0.06	0.49
	6	1.40 ± 0.11	0.40
	10	1.96 ± 0.15	0.33
+ MSG	0	1.04 ± 0.02	0.51
	10	1.12 ± 0.05	0.55
	30	1.06 ± 0.03	0.54
	80	2.13 ± 0.05	0.49
	100	8.83 ± 0.09	0.47
+ IMP	0	0.92 ± 0.03	0.54
	1	0.97 ± 0.04	0.53
	10	1.73 ± 0.06	0.53
	20	1.45 ± 0.13	0.40
	30	1.95 ± 0.09	0.27
+ l-Theanine	0	0.89 ± 0.05	0.56
	10	0.78 ± 0.06	0.53
	30	1.00 ± 0.04	0.50
	80	1.65 ± 0.14	0.43
	100	2.74 ± 0.17	0.31

^†^ Means ± SEM (all such values), ^‡^ NE: not estimated.

**Table 2 ijms-21-07045-t002:** Changes of IC_50_ values of Glu-Glu, MSG, and l-theanine in the presence of 10 mM salicin by adding graded concentration of IMP in cells expressing hTAS2R16.

10 mM Salicin + Sample	+ IMP (mM)	IC_50_ (mM)
+ Glu-Glu	0	7.25 ± 0.02
	1	7.03 ± 0.01
	3	6.97 ± 0.02
	10	7.28 ± 0.02
+ MSG	0	64.67 ± 0.04
	1	59.85 ± 0.05
	5	49.65 ± 0.06
	10	19.51 ± 0.09
+ l-Theanine	0	77.33 ± 0.05
	1	78.13 ± 0.06
	3	79.86 ± 0.06
	10	69.62 ± 0.06

Means ± SEM (all such values).

**Table 3 ijms-21-07045-t003:** Changes of EC_50_ values and maximal responses of salicin by pretreatment of umami substances in wild-type (WT) and two known probenecid-insensitive mutants.

Sample (mM)		EC_50_ (mM)			Maximal Response (ΔF/F_0_)		
		WT	N96T	P44T	WT	N96T	P44T
+ Probenecid	0	0.90 ± 0.04 ^†^	0.23 ± 0.03	0.64 ± 0.06	0.54	0.68	0.56
	0.3	1.04 ± 0.19	0.26 ± 0.03	0.94 ± 0.10	0.23	0.68	0.58
+ Glu-Glu	0	1.04 ± 0.03	0.25 ± 0.05	0.62 ± 0.04	0.54	0.61	0.62
	6	1.35 ± 0.12	0.24 ± 0.07	0.59 ± 0.07	0.40	0.65	0.42
+ MSG	0	1.07 ± 0.02	0.18 ± 0.04	0.68 ± 0.03	0.51	0.66	0.56
	80	2.03 ± 0.06	0.29 ± 0.10	0.94 ± 0.14	0.49	0.65	0.57
+ IMP	0	0.96 ± 0.03	0.20 ± 0.03	0.69 ± 0.04	0.54	0.66	0.61
	20	0.97 ± 0.12	0.30 ± 0.15	0.94 ± 0.13	0.40	0.28	0.37
+ l-Theanine	0	1.10 ± 0.03	0.20 ± 0.04	0.75 ± 0.04	0.56	0.63	0.59
	80	1.73 ± 0.16	0.31 ± 0.07	1.03 ± 0.03	0.43	0.66	0.56

^†^ Means ± SEM (all such values).

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
