# Peer review of "Suppression of hTAS2R16 Signaling by Umami Substances"

_ijms, 2020, doi:10.3390/ijms21197045_

Round 1

Reviewer 1 Report

1. The sentences in lines 30-31; 55-57; 62-65; 151-153 require editing for better and clearer understanding.

2. The name of gene TAS2R16 in key words should be in italics.

3. References 20 is not correctly written and should be edited.

4. Providing the clinical context of the study might be considered by the Authors 

Author Response

  1. The sentences in lines 30-31; 55-57; 62-65; 151-153 require editing for better and clearer understanding.

Response: According to the reviewer’s comment, we edited whole of “Introduction” including the sentences in lines 30-31; 55-57; 62-65, and we also edited 151-153 for better understanding.

  1. The name of gene TAS2R16 in key words should be in italics.

Response: We changed it into italic, thanks.

  1. Rerences 20 is not correctly written and should be edited.

Response: We changed the title based on original manuscript, thanks.

  1. Providing the clinical context of the study might be considered by the Authors 

Response: Thank you for this pertinent comment. As you know, taste receptors for sweet, umami, and bitter tastes, are actually expressed in the extra oral tissues, and those receptors have been regarded as potentially important therapeutic targets. The sentences were added in the Introduction section, lines 89-95, and we added matching reference as a number 42 (a recent review by Jeruzal-ÅšwiÄ…tecka, J. et al., Int. J. Mol. Sci., 21, 5156 (2020)).

Lines 89-95: Recently, it has been elucidated that bitter taste receptors are also actually expressed in the extra oral tissues, such as, intestine, brain, bladder, and lower and upper respiratory tract. Furthermore, bitter taste receptors have been identified as potentially important therapeutic targets for diseases associated with the above extra oral tissues. The detailed molecular mechanisms involved in regulating the activity of bitter taste receptor, therefore, can provide useful information for the future design and development of therapeutic intervention.

Reviewer 2 Report

In the current work authors characterized “Suppression of hT2R16 by umami substances”. Authors have done significant amount of work and presented their work nicely. However, author needs to justify these following comments for the paper.

General comments:

  1. All the umami compounds showed to block the T2R16 response were used at very concentration (high mM range), what is the justification for this?

  1. As we know that umami and bitter receptors upon activation can signal through cAMP not calcium always, so at least with one umami agonist authors should show data for cAMP production for the inhibiton assay.

3.On page 5, line 127, 10mM of salicin concentration is mentioned, is it the EC50 value for T2R16?

  1. Authors should provide raw calcium traces for at least one umami compound showing blocking of T2R16.

Author Response

  1. All the umami compounds showed to block the T2R16 response were used at very concentration (high mM range), what is the justification for this. Response: There are many cases in which our ordinary foods contain large amounts of umami substances such as glutamate. For example, tomato juice contains 150-250 mg glutamate/100 g fluid, and it is estimated to be equivalent to approximately 10-17 mM. In addition, the threshold of human umami receptor against MSG has been reported as about 3 mM. Taken together, it is not surprising that umami substances were used at high concentrations (mM range) to estimate the event occurring in our oral cavity. Thank you for your kind consideration.
  2. As we know that umami and bitter receptors upon activation can signal through cAMP not calcium always, so at least with one umami agonist authors should show data for cAMP production for the inhibition assay. Response: Of course, we also understand the presence of the signaling pathway through regulating intracellular cAMP level for bitter taste in the taste receptor cells. However, in the heterologous expression system for taste GPCRs including bitter taste receptors, researchers ordinarily utilized the measurement of cellular Ca2+ responses in HEK293 cells that were expressing the bitter taste receptor together with a chimeric G-protein (such as G16αgust44) to accomplish the sensitive analysis. There are many publications regarding the functional characterization for bitter taste receptors using this assay system at present, it should be a reliable methodology for the receptors. Thanks for your detailed consideration
  3. On page 5, line 127, 10mM of salicin concentration is mentioned, is it the EC50 value for T2R16?.                                                              Response: 10 mM of salicin is not a EC50 value, this is the concentration of salicin approaches a plateau that is fully induced level. Because we were intended to observe inhibition of binding of salicin by umami substances, we used this concentration in this study.
  4. Authors should provide raw calcium traces for at least one umami compound showing blocking of T2R16.                                                        Response: This is very appropriate point to give credibility of the data. We added raw calcium traces for MSG, Glu-Glu, IMP and L-theanine in Figure 1 as B. For this reason, Figure 1B, C, and D was changed to Figure 1C, D, and E. We also added legend for Figure 1B.